# High-Strength, Degradable and Recyclable Epoxy Resin Based on Imine Bonds for Its Carbon-Fiber-Reinforced Composites

**DOI:** 10.3390/ma16041604

**Published:** 2023-02-15

**Authors:** Yue Jiang, Shuai Wang, Weifu Dong, Tatsuo Kaneko, Mingqing Chen, Dongjian Shi

**Affiliations:** 1Key Laboratory of Synthetic and Biological Colloids, Ministry of Education, School of Chemical and Material Engineering, Jiangnan University, Wuxi 214122, China; 2Graduate School of Advanced Science and Technology, Japan Advanced Institute of Science and Technology, Nomi 923-1292, Ishikawa, Japan

**Keywords:** epoxy resin, degradable, recycled, bio-based

## Abstract

Carbon fiber (CF) is widely used in the preparation of carbon-fiber-reinforced polymer composites (CFRP) in which it is combined with epoxy resin due to its good mechanical properties. Thermosetting bisphenol A epoxy resin, as one of the most common polymer materials, is a non-renewable resource, leading to a heavy environmental burden and resource waste. To solve the above problems and achieve high mechanical and thermal properties comparable to those of bisphenol A, herein, a high-performance, degradable and recyclable bio-based epoxy resin was developed by reacting the lignin derivative vanillin with 4-amino cyclohexanol via Schiff base. This bio-based epoxy resin showed a Young’s modulus of 2.68 GPa and tensile strength of 44 MPa, 36.8% and 15.8% higher than those of bisphenol A epoxy, respectively. Based on the reversible exchange reaction of the imine bond, the resin exhibited good degradation in an acidic environment and was recoverable by heat treatment. Moreover, the prepared epoxy resin could be used to prepare carbon fiber (CF)-reinforced composites. By washing off the epoxy resin, the carbon fiber could be completely recycled. The recovered carbon fiber was well preserved and could be used again for the preparation of composite materials to realize the complete recovery and utilization of carbon fiber. This study opens a way for the preparation of high-performance epoxy resin and the effective recycling of carbon fiber.

## 1. Introduction

Carbon-fiber-reinforced polymer composites (CFRP) are widely used in aerospace, wind turbine blades, automobile parts and other fields due to their advantages of high modulus, high strength, low density and corrosion resistance [1,2,3,4]. The polymer substrates in CFRP include epoxy resins, dual maleimide resins and polyimide resins. Epoxy resins are some of the most significant thermosetting polymers, contributing good thermal and mechanical properties, chemical resistance and adhesion [1]. However, more than 90 percent of thermosetting epoxy resins are prepared from bisphenol A glycidyl ether (DGEBA) [5,6]. These traditional DGEBA epoxy resins are derived from fossil resources. The permanent cross-linked networks formed within the resins make them non-degradable and non-reprocessing composites. Recycling by landfill incineration causes a huge burden on the environment. These methods may reduce the quality of carbon fiber and greatly reduce its commercial value [7,8,9,10]. Therefore, finding how to gently recover waste thermosetting epoxy resin and then recover the high-cost carbon fiber has become a primary problem.

The large number of three-dimensional networks in CFRP makes them composite materials with high stability, which is the fundamental factor hindering the recovery of CFRP composite materials. The key to solving this problem is to change the stable cross-linking structure in the material [11,12]. A covalent adaptive network (CAN) is one effective structure to solve this problem [13]. The reversible dynamic covalent bonds include the ester bond, [14,15,16] acetal bond, imine bond, [17,18,19] disulfide bond, [20,21,22,23] ester bond and Diels–Alder (DA) bond [24,25,26,27]. The network topologies of the resins with the above dynamic covalent bonds were rearranged through a change in the external environment, providing thermosetting resins with a certain plasticity [17,19,28]. In order to achieve a high exchange rate of the reversible bonds, catalysts are generally required during the preparation of a CAN with the polymer chains, except in the case of imine bonds. The CAN could be recycled into soluble small molecules through mild stimulation, which has a certain potential to replace traditional thermosetting resin in the preparation of CFRP composites. 

Improving the recovery efficiency of thermosetting resins and CFRP composites has become an urgent problem. In order to replace traditional thermosetting resins, in this paper, we aim to fabricate a bio-based epoxy resin with good mechanical properties comparable to those of DGEBA, as well as with good degradability and easy recoverability without any other catalysts. Vanillin is the only commercially available lignin aromatic derivative, and it is a renewable resource with high structural design ability, so it has attracted extensive attention in the polymer field. Herein, a novel bio-based monomer (VAN-AC) composed of vanillin (a lignin derivative) and 4-aminocyclohexanol was prepared via Schiff base reaction. After reacting the VAN-AC with epichlorohydrin (ECH), an epoxy resin (VAN-AC-EP) was obtained and further thermally cured with the commercial curing agent 4,4′-diaminodiphenylmethane (DDM). The curing kinetics, mechanical properties, thermodynamic properties, degradability and recoverability of VAN-AC-EP/DDM were systematically studied and compared with those of DGEBA/DDM. On this basis, VAN-AC-EP CFRP composites were further prepared by adding carbon fiber (CF) into the VAN-AC-EP, and the mechanical properties of the CFRP composites and the recovery of the CF were investigated.

## 2. Experimental Part

### 2.1. Materials

Vanillin (VAN), 4-amino-cyclohexanol (AC), 4,4-diamino-diphenylmethane (DDM), tetrabutylammonium bromide (TBAB) and potassium hydroxide (KOH) were purchased from Aladdin Corporation, China. Ethanol, epichlorohydrin (ECH), methylene chloride, hydrochloric acid (HCl) and ethylenediamine were purchased from Sinophosphoric Chemical Reagent Co, Ltd., Shanghai China. Diglycidyl ether of bisphenol A (DGEBA) epoxy resin was supplied by McLean, China. Carbon fiber (T300-3k, 240 g/m^2^) was purchased locally. All compounds were directly used without any treatments, unless otherwise stated.

### 2.2. Synthesis of Vanillin-Based Epoxy Monomer (VAN-AC-EP)

As shown in Figure 1, the synthesis of vanillin-based epoxy monomer (VAN-AC-EP) was divided into two steps. First, as shown in Figure 1, 45.6 g (0.3 mol) vanillin and 34.5 g (0.3 mol) 4-aminocyclohexanol were dissolved in 200 mL of ethanol with stirring, and the mixture was refluxed at 80 °C for 2 h. The yellow solid that precipitated from the solution was filtered and washed with ethanol three times. After drying in an oven at 50 °C for 1 h, a yellowish powder (VAN-AC) was obtained. The yield was 84.2%.

In the second step, 24.9 g (0.1 mol) VAN-AC powder was dissolved in 92.5 g (1 mol) epichlorohydrin, and 1.6 g TBAB was added into the mixture. After stirring at 120 °C for 24 h, the mixture was rapidly cooled to 0 °C by placing the reaction flask in an ice water bath. Then, 60 g KOH aqueous solution (50 wt%) was added dropwise, and the reaction was carried out for another 2 h. At the end of the reaction, a stratified system was obtained. After separation, the underlying oil was diluted by dichloromethane, washed with deionized water three times, and rotary evaporated to obtain a yellow liquid (VAN-AC-EP). The yield was 81.4%.

### 2.3. Preparation of Thermosetting Epoxy Resin (VAN-AC-EP/DDM)

Quantities of 10.8 g VAN-AC-EP and 3 g DDM (1:1 molar ratio of epoxy groups to N-H groups) were heated to melting and stirred well in a beaker. After cooling at room temperature, the mixture was debubbled in a vacuum oven for 10 min and then poured into a mold. Then, it was heated at 100 °C for 3 h and subsequently cured at 110 °C, 130 °C and 150 °C for 2 h respectively. After cooling at room temperature, the thermosetting resin VAN-AC-EP/DDM was demolded. As comparison samples, DGEBA and DDM were also cured in accordance with the above ratio.

### 2.4. Reprocessing of Thermosetting Epoxy Resin (VAN-AC-EP/DDM)

VAN-AC-EP was ground into a powder, evenly spread in a steel groove 70 mm (L) × 5 mm (W) × 0.5 mm (T) with a polytetrafluoroethylene plate above and below, and placed in a plate vulcanizing machine for hot pressing at 180 °C for 0.5 h. Recycled film was obtained by demolding after cooling at room temperature.

### 2.5. Preparation of Carbon-Fiber-Reinforced Polymer Composites (CFRP)

Firstly, VAN-AC-EP and DDM were fully melted and mixed at 90 °C, and 50 wt% ethanol was added to make a mixed solution. The mixture was evenly brushed onto carbon fiber (100 mm × 100 mm). Then, carbon fiber prepreg was obtained by removing the ethanol in a vacuum oven at 60 °C for 10 h. The carbon fiber composites were prepared by hot pressing at 110 °C, 130 °C and 150 °C for 2 h respectively 

### 2.6. Reprocessing of Carbon-Fiber-Reinforced Polymer Composites (CFRP)

Broken parts of CFRP were overlapped and sandwiched between two polytetrafluoroethyene plates. They were then hot pressed at 180 °C for 0.5 h. After cooling, recycled CFRP was obtained.

### 2.7. Gel Fraction Measurement

The gel fraction of VAN-AC-EP/DDM was determined by the impregnation method. The sample with original mass M_0_ was immersed in several common solvents and placed at 50 °C for 24 h. It was then removed from the solvent, and the mass M_s_ after the removal of excess liquid on the surface was recorded. Then, the sample was dried in a drying oven under vacuum. After the mass of the sample reached a constant, the mass was recorded as M_d_, and the swelling rate and gel content were calculated as follows (1)–(2) [29]:(1)$$\mathrm{Swell}\;\mathrm{ratio}=(m_{s}-m_{o})/m_{o}\times 100\%$$
(2)$$\mathrm{Gel}\;\mathrm{content}=m_{d}/m_{o}\times 100\%$$

### 2.8. Characterization

Proton nuclear magnetic resonance spectra (^1^H NMR) were recorded on an AVANCE III Bruker NMR spectrometer (Bruker, Fällanden, Switzerland) using DMSO-d6 as a solvent. Fourier transform infrared spectra (FT-IR) were recorded using a SPECTRUM 100 spectrophotometer (Perkin Elmer, Waltham, MA, USA) from 4000 to 600 cm^−1^ with a resolution of 4 cm^−1^ and a scanning number of 32 times. Differential scanning calorimetry (DSC) curves were recorded using a NETZSCH DSC-214 (Selb, Germany). Measurements were carried out at a rate of 10 °C/min at 50–250 °C under N_2_ atmosphere. The tensile properties of VAN-AC-EP-DDM and CFRP at room temperature were tested on a 5967X universal testing machine. The size of the resin samples for tensile testing was 30 mm (L) × 5 mm (W) × 2 mm (T), and the stretching rate was 10 mm/min. The distance between the two clamps was 10 mm. Each tensile value was obtained from an average of 5 samples. Dynamic Mechanical Analysis (DMA) curves were recorded using a DMA Q800 under tensile mode (TA Instruments, New Castle, DE, USA). Test samples of size 30 mm (L) × 5 mm (W) × 2 mm (T) were prepared, and the measurement was carried out between 30 and 250 °C. The heating rate was 3 °C/min, and the oscillation frequency was 1 Hz. The stress relaxation property was determined as follows: samples were equilibrated between machine clamps for 5 min at a pre-set temperature, and then 1% strain was applied to the sample. Changes in the storage modulus over time were recorded for analysis. Thermogravimetric analysis (TGA) was performed on a TGA/1100SF device at temperatures ranging from 50 to 800 °C and at a heating rate of 10 °C/min in a nitrogen atmosphere. The morphologies of the carbon residue after TGA measurements and of the combination of carbon fiber and resin were investigated by scanning electron microscopy (SEM, S-4800) at 2.0 kV. Raman spectra were measured using an inVia Reflex laser with a laser wavelength of 532 nm, an exposure time of 60 s and a laser energy of 12 mW. 

## 3. Results and Discussion

The molecular structures of VAN-AC and VAN-AC-EP were determined from their ^1^H NMR and FT-IR spectra. In Figure 1a, peaks at 9.43 ppm (s, 1H, -CH=N-), 8.20 ppm (s, 1H, -Ar-OH), 7.28 ppm (dd, 1H, ArH), 7.08 ppm (dd, 1H, ArH), 6.81 ppm (dd, 1H, ArH), 4.55 ppm (s, 1H, -OH), 3.78 ppm (m, 3H, -O-CH_3_), 3.45 ppm (s, 1H, -CH-), 3.10 ppm (s, 1H, -CH-), 1.87–1.56 ppm (dd, 4H), 1.57–1.27 ppm (dd, 4H, -CH_2_-) and 3.39–3.33 ppm (m, 1H, -CH_2_-) were assigned to the protons of VAN-AC. Figure 1b shows the ^1^H NMR spectrum of VAN-AC-EP. The characteristic peaks belonging to the hydroxyl protons at δ = 8.20 and 4.55 disappeared, and new characteristic peaks appeared at 4.44, 3.94, 1.79, 1.61, 1.23 and 1.09 ppm, which were attributed to the proton peak of the glycidyl ether. The FT-IR spectra of VAN-AC and VAN-AC-EP are shown in Figure 1c. The disappearance of the hydroxyl characteristic peak at 3300 cm^−1^ and the appearance of the epoxy characteristic peak at 910 cm^−1^ also proved the successful synthesis of the VAN-AC-EP epoxy resin. 

The curing kinetics of the VAN-AC-EP/DDM system were studied by differential scanning calorimetry (DSC) at four heating rates of 5, 10, 15 and 20 °C/min. Figure 2a shows the non-isothermal DSC curves of VAN-AC-EP/DDM. An obvious exothermic peak can be seen. By integrating the exothermic peak, the exothermic enthalpy (ΔH), initial curing temperature (T_b_), peak curing temperature (T_p_) and end curing temperature (T_e_) could be obtained. As a comparison, the same conditions were used to test DGEBA/DDM. For both epoxy resins, the faster the heating rate was, the higher the curing temperature was. The solidification activation energy (E_a_) can be calculated according to Kissinger formula (3) [30]:(3)$$\ln({\beta /\mathrm{Tp}}^{2})=\ln(\mathrm{AR}/E_{a})-E_{a}/{RT}_{p}$$
where β is the heating rate, A is the preexponential factor and R is the ideal gas constant. By linear fitting, ln(β/T_p_^2^) plotted against 1/T_p_ was obtained (Appendix A), and the activation energy was calculated from the slope. The data are shown in Table 1. It can be clearly seen from the table that the initial curing temperature (T_b_), peak curing temperature (T_p_) and end curing temperature (T_e_) of the VAN-AC-EP/DDM system were lower than those of the DGEBA/DDM system. The enthalpy of release (ΔH) of the VAN-AC-EP/DDM system was much lower than that of the DGEBA/DDM system. The curing process of VAN-AC-EP/DDM was faster and offers greater environmental protection in regard to energy savings. However, the curing activation energy was higher than that of DGEBA/DDM, possibly due to the high content of C=N double bonds in the main chain of the VAN-AC-EP/DDM system inducing an increase in steric hindrance [30]; this is not conducive to diffusion of the epoxy monomer and curing agent. 

The thermal properties of the VAN-AC-EP/DDM and DGEBA/DDM systems were characterized by TGA under N_2_ atmosphere. As shown in Figure 3a, the initial thermal decomposition temperature (5% mass reduction) of the VAN-AC-EP/DDM system was 268 °C, which was lower than that of the DGEBA/DDM system (372 °C). This is induced by the molecular structure of VAN-AC-EP/DDM, wherein the C=N double bond is unstable and easily decomposes at high temperature. However, the temperature of 268 °C is high enough to cope with most applications of thermosetting resin. The thermal stability of a polymer is usually represented by its heat resistance index (T_s_), which can be calculated via Equation (4) [29]:(4)$$T_{s}=0{.49[T}_{d5\%}+0{.6(T}_{d30\%}{-T}_{d5\%})]T$$
where T_d5%_ is the temperature at which the mass is reduced by 5% and T_d30%_ is the temperature at which the mass is reduced by 30%. The thermal performance is shown in Table 2.

As shown in Table 2, the heat resistance index of the VAN-AC-EP/DDM system (146 °C) was lower than that of the DGEBA/DDM system (184 °C). The carbon residue rate of the VAN-AC-EP/DDM system (25.8%) was higher than that of the DGEBA/DDM system (16.3%) [31,32]. This higher carbon residue rate marks VAN-AC-EP/DDM for potential applications in flame retardants. It could be seen more clearly from the DTG curve (Figure 3b) that the pyrolysis of the imine bond corresponded to the first mass loss peak in the DTG curve. In the subsequent pyrolysis, the mass loss peak of the VAN-AC-EP/DDM system was much lower than that of the DGEBA/DDM system. This was because the C=N double bond underwent the amine exchange reaction at a high temperature; further, the aromatic imine bond has a corresponding self-crosslinking ability, forming a nitrogenous six-membered ring at high temperature and then transforming into a dense carbon layer, reducing the pyrolysis rate. The carbon residue of the DGEBA/DDM system was full of holes, indicating that the internal and external thermal decomposition was consistent. On the contrary, the carbon residue of the VAN-AC-EP/DDM system had a smooth surface without pores, and the internal carbon base was protected, [33] as confirmed by the scanning electron microscopy (SEM) images in Appendix A.

The stress–strain curves showed that the tensile strength of VAN-AC-EP/DDM reached 44 MPa, higher than that of DGEBA/DDM (38 MPa), while the elongation at break was 2.4%, slightly lower than that of the DGEBA/DDM system, and the Young’s modulus of VAN-AC-EP/DDM was 2685 MPa, much higher than that of DGEBA/DDM (2320 MPa). The higher mechanical strength of VAN-AC-EP/DDM is provided by the large number of C=N double bonds in the molecular backbone of the VAN-AC-EP/DDM system after curing and the π–π conjugation present between the benzene ring and the imine bond. The higher molecular rigidity limited the rotation of the molecular chains, which induced the decrement in the flexibility.

The thermodynamic properties of VAN-AC-EP/DDM and DGEBA/DDM were characterized by DMA. As an important parameter of the upper limit temperature of thermosetting resin, the glass transition temperature (T_g_) of VAN-AC-EP/DDM (146 °C) was lower than that of the DGEBA/DDM system (201 °C), according to the Tan Delta curve in Figure 3d. The energy storage modulus of VAN-AC-EP/DDM was up to 3350 MPa, much higher than that of DGEBA/DDM (2320 MPa). VAN-AC-EP and DGEBA have similar structure, but the energy storage modulus of VAN-AC-EP/DDM decreases more rapidly due to the relatively lower heat resistance of the aliphatic ring in VAN-AC-EP than that of the benzene ring in DGEBA. 

The cross-linking densities of VAN-AC-EP/DDM and DGEBA/DDM can be calculated as their rubber elasticity via Equation (5) [34]:(5)$$\mathrm{Er}=3RT_{r}V_{e}$$
where T_r_ = T_g_ + 40K, Er is the energy storage modulus at T_r_, V_e_ is the cross-linking density and R is the gas constant (8.314 Jmol^−1^K^−1^). The data are shown in Appendix A. The cross-linking density of VAN-AC-EP/DDM was 568 mol/m^3^, lower than that of DGEBA/DDM. Generally, a system with higher cross-linking density leads to higher energy storage modulus and T_g_. T_g_ conforms to this rule, but the energy storage modulus goes against it. This is because a large number of imine bonds exist in the molecular backbone; thus, the network structure of VAN-AC-EP/DDM is relatively harder, and the stiffness of VAN-AC-EP/DDM is nearly twice that of DGEBA/DDM. 

The ductility of VAN-AC-EP/DDM was tested by stress relaxation experiments. In Figure 4a, four temperatures of 160, 170, 180 and 190 °C were selected for testing. Due to the large number of dynamic covalent bonds in VAN-AC-EP/DDM, an obvious relaxation phenomenon occurred. The higher the temperature was, the more obvious the stress relaxation phenomenon was, fully showing the temperature dependence. When the temperature was higher than T_g_, the relaxation time τ followed Arrhenius’ law [35]:(6)$$\tau *(T)\;=\tau o\;\exp\;(E_{a}/\mathrm{RT})$$
where τ* is the time required for the modulus to reach 1/e of the initial modulus, τ_0_ is the characteristic relaxation time at infinite temperature, E_a_ is the activation energy of bond exchange, R is the universal gas constant (8.314 jmol^−1^k^−1^) and T is the temperature in units of K. Linear fitting was performed by the curve of ln(τ*) plotted against 1000/T (Figure 4b), and E_a_ (82.3 KJ/mol) was calculated from the slope. Apparent relaxation of the resin occurred due to the rearrangement of imine bonds. The dynamic exchange of amine bonds provided the reprocessing ability of the thermosetting resin. It only took 2 min for the stress of the material to drop to 1/e of the initial value at 190 °C. The value of Ea is noteworthy, as it is much lower than those of some other thermosetting epoxy resins without the addition of catalysts. This makes the resin more malleable at high temperature.

Since VAN-AC-EP/DDM had good ductility, its reprocessed performance could be explored. The cured VAN-AC-EP/DDM was ground into a powder, and resin film was obtained by hot pressing. Figure 5a shows that the composition of the recycled VAN-AC-EP/DDM did not change compared with that of the original VAN-AC-EP/DDM. The stretching vibration peak (1580 cm^−1^) belonging to C=N was also retained. From the tensile curves in Figure 5b, the elongation at break of the recovered sample was 1.1%, 45.8% of the original value. Its tensile strength was 29 MPa, 65.9% of the initial value. The Young’s modulus of the recycled resin was 2979 MPa, which was almost unchanged and even higher than the original value (2685 MPa), because the cross-linking density of the reprocessed thermosetting resin became higher. It was inevitable that the reprocessed resin was lower in performance than the original resin. The C=N double bonds under high temperature and high pressure could be reversible by amine exchange recovery, while other covalent bonds were irreversible and damaged. This might be one of the reasons for the degradation. In addition, the resin presented an aging-resistance phenomenon after a long period of exposure to high-temperature conditions.

VAN-AC-EP/DDM was expected to have hydrolyzed degradation ability in an acidic environment due to the imine bonds. The process of degradation was divided into two parts. Firstly, the thermosetting resin expanded in the degradation solution, and then the polymer chains dissociated. Therefore, the degradation rate of the thermosetting resin was mainly affected by the infiltration and hydrolysis rate. Generally speaking, the degradation solution should be in full contact with the thermosetting resin and spread into its interior. The contact angle (θ) is an important parameter for the infiltration of liquid into a solid [30]. The smaller θ is, the better the infiltration is. VAN-AC-EP/DDM in pure HCl solution was not very infiltrative (θ = 66.4°). However, in acetone, it had good swelling infiltration (θ = 16.7°), as shown in Table 3. Because acetone is volatile, the mixture of acetone and 1 mol/L HCL solution was chosen as a degradation solvent. The contact angle test was performed for different ratios of acetone and hydrochloric acid mixtures, as shown in Appendix A. In order to balance infiltration and acidity (Appendix A), the solution with an acetone/HCL volume ratio of 5:5 was finally configured as the degradation solution.

The degradation process is shown in Figure 6. The thermosetting resin was completely degraded in 4 h. A comparative experiment was also made, as shown in Appendix A. After VAN-AC-EP/DDM was put into acetic acid for 4 h, there was no degradation behavior to be found.

VAN-AC-EP/DDM was immersed in ethylenediamine, and the degradation process is shown in Figure 6c. VAN-AC-EP/DDM degraded in acidic solvent by the breaking of covalent bonds, while in amine solution, it performed an amine exchange reaction with small active amine molecules. Through this amine exchange reaction, resin with molecular weight in the hundreds and thousands was converted into small molecules. As also reflected in the digital photos, VAN-AC-EP/DDM first degraded into small pieces in amine solution and was then completely digested by ethylenediamine.

The components of the degradation products were analyzed by examining the ^1^H NMR and FT-IR spectra. In Figure 6e, the product degraded in ethylenediamine still retained the characteristic peak (9.5 ppm) of the imine bond, and the stretching vibration peak (1580 cm^−1^) belonging to C=N was also retained in Figure 6f. However, for the products degraded in an acidic environment, no signal of the imine bonds was found in the ^1^H NMR spectrum, and a stretching vibration peak belonging to C=O (1660 cm^−1^) appeared in FT-IR (Figure 6f), which also indicated that the degradation process was carried out according to the mechanism.

VAN-AC-EP/DDM was immersed in several common polar solvents at 50 °C for 12 h to characterize the chemical stability of the thermosetting resin. The specific data are shown in Table 2. VAN-AC-EP/DDM remained stable without mass loss after immersion in strong polar solvents for a long time. It showed a low swelling rate and a high gel fraction, which was attributed to the dense cross-linking network of thermosetting resin, high degree of curing and excellent chemical stability. However, the aging of the resin would be accelerated in a high-humidity environment. The dampness and heat resistance stability has become an important index used to evaluate the stability of thermosetting resin, especially hydrolyzable thermosetting resin. In Figure 7, the T_g_ of VAN-AC-EP/DDM only decreased by 5 °C for 20 d in a high-temperature and -humidity environment (70% humidity, 60 °C).

The carbon-fiber-reinforced composites were prepared according to the procedures shown in Figure 8a,b. The mechanical properties of the composites were characterized by tensile experiments. As shown in Figure 9a, the initial elongation at break reached 6.1%, and the tensile stress reached 122 MPa. The VAN-AC-EP/DDM on the carbon fiber was washed away easily by immersing the composite material in HCl solution, and then the carbon fiber was successfully recovered by simply cleaning and drying the carbon fiber with deionized water. Compared with the original carbon fiber (Figure 9c), through SEM observation, the recycled carbon fiber surface was smooth and without redundant VAN-AC-EP/DDM residues (Figure 9d). In the Raman spectra (Figure 9b) the chemical structure of recycled CFRP was consistent with that of the original CFRP. All these phenomena indicate that the carbon fiber composites could be recovered without damaging the carbon fiber structure, providing energy savings and environmental friendliness.

Bio-based epoxy resin has the characteristics of rapid curing, being already hardened when melted and mixed with a curing agent. It is thus difficult to produce finished products by pouring at room temperature. Most of the bio-based resins are prepared by pressing sheets, which greatly limits their application. SEM showed that the carbon fiber was completely encapsulated by VAN-AC-EP/DDM, rather than forming a thermosetting resin film on the surface (Figure 9e), which was the reason for the good mechanical properties of the carbon fiber composite. The carbon fiber composite could also be reprocessed, and the tensile strength (80 MPa) and strain (2%) of the reprocessed carbon fiber composite could be restored to 67% and 34% those of the original CFRP, respectively.

## 4. Conclusions

In summary, a novel epoxy resin, VAN-AC-EP, with recyclable and recoverable properties was synthesized using a bio-based vanillin monomer. This bio-based resin had good thermodynamic properties after thermal curing. The T_g_ value was 146 °C; in addition, the tensile strength was 44 MPa, and the Young’s modulus was 2.68 GPa, which were 15.8% and 36.8% higher than those of bisphenol A epoxy, respectively. In addition, the dynamic imine bond introduced by the Schiff base reaction gave the resin degradation and recycling ability. It could release deformation stress quickly at 170 °C under high temperature, and it had good reprocessing ability. Using the epoxy resin as the matrix, the resulting carbon-fiber-reinforced composite had the same excellent mechanical properties, and the tensile strength was 122 MPa. After removing the epoxy matrix efficiently, the carbon fiber could be fully recovered, and the carbon fiber could still be used in composite materials. Due to the low cost of recycling resin and carbon fiber, there is hope that this bio-based epoxy resin (VAN-AC-EP) and its CFRP composite could have considerable application prospects in wind power generation and aerospace fields.

## Data Availability

The data presented in this study are available on request from the corresponding author. The data are not publicly available.

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
