# Peer review of "High-Strength, Degradable and Recyclable Epoxy Resin Based on Imine Bonds for Its Carbon-Fiber-Reinforced Composites"

_materials, 2023, doi:10.3390/ma16041604_

Round 1

Reviewer 1 Report

1. There is little information about Vanillin. It is important to describe the source material of the resin better.

2. Figure 1 has a low image quality; it is impossible to distinguish the peaks from H NMR as well as the FTIR spectra.

3. The caption of Figure 1 is wrong.

4. On page 5, sentence: “It could be clearly seen from the table that the initial curing temperature (Tb), peak curing temperature (Tp) and end curing temperature (Te) of VAN-AC-EP/DDM system were higher than 30 °C lower than those of DGEBA/DDM system”.
This claim is confusing and does not agree with the results.

5. On page 6, when discussing the heat resistance index (Ts), it is necessary to add references that demonstrate the claim.

6. The Equation and Ts values should be in the article and not as supplementary material, as it is an important discussion of the work.

7. The calculated value for Ts of the DGEBA is wrong. Consequently, the statement that VAN-AC-EP/DDM has greater thermal stability than DGEBA/DDM is invalid.

8. Page 6, sentence: “The higher carbon residue rate marked the VAN-AC-EP/DDM have a potential application in flame retardant.”
It is necessary to add references that demonstrate the claim.

9. Figure 3d should differentiate the E' and Tan delta curves.

10. Table S2 is titled mechanical properties, but the first result is the Tg of the materials. Furthermore, the Tg value in Table S2 differs from the values reported in Table S1. Why are they different values?

11. How were the Stiffness (kN/m) values shown in Table S2 obtained?

12. What is the experimental error of tensile tests?

13. Later in the article, the elastic modulus value is mentioned. Are these values obtained from Figure 3c? There is a shoulder at the beginning of the DGEBA sample tensile test curve. What does this behavior refer to, and how was the elastic modulus considered?

14. On page 7, in the text "Equation 2" the number is incorrect.

15. The values of Tr in Table S2 are wrong. And apparently, the correct Er region of the DGEBA sample is not plotted on the DMA plot (Figure 3d).

16. On page 8, there is mention of Figure S3. However, there is no Figure S3. Possibly this is Figure 4b.

17. On page 8, the third sentence statement needs further discussion and references.

18. Figure 5a needs to discuss the main vibrations of interest and demonstrate them graphically in the spectra.

19. What are the dimensions of the recycled VAN-AC-EP/DDM specimen for tensile testing, and what is the experimental error of the analysis?

20. Figure S4 does not have a volume ratio of the acetone/HCL scale.

21. Figure 7, what are the legends of the DSC curves?

22. In the conclusions, a third Tg value is reported, different from the other two in Tables S1 and S2. Why the different values? The value of the modulus of elasticity was not discussed throughout the text.

Author Response

Response to Reviewers

Thank you very much for your valuable comments. We have revised the manuscript thoroughly according to the reviewers’ recommendations.

For Reviewer #1:

Point1: There is little information about Vanillin. It is important to describe the source material of the resin better.

Response: Thank you for your comments.The sentence has been added Vanillin is the only commercially available lignin aromatic derivatives, which is a renewbale resource and has a high structural design ability,so it has attracted extensive attention in the polymer field.

Point2: Figure 1 has a low image quality; it is impossible to distinguish the peaks from H NMR as well as the FTIR spectra.

Response: Thanks for the reviewer’s suggestion. The figure has been reuploaded.

Point3: The caption of Figure 1 is wrong.

Response: We have modified the caption in the article Figure 1. 1H NMR spectra of VAN-AC (a) and VAN-AC-EP (b),FT-IR spectra of VAN-AC and VAN-AC-EP(c).

Point4: On page 5, sentence: “It could be clearly seen from the table that the initial curing temperature (Tb), peak curing temperature (Tp) and end curing temperature (Te) of VAN-AC-EP/DDM system were higher than 30 °C lower than those of DGEBA/DDM system”.

This claim is confusing and does not agree with the results.

Response: Most of them are below 30℃ or above. The sentense is not strict enough. The sentence has been amended. The data are shown in Table 1. It could be clearly seen from the table that the initial curing temperature (Tb), peak curing temperature (Tp) and end curing temperature (Te) of VAN-AC-EP/DDM system were lower than those of DGEBA/DDM system.

Point5: On page 6, when discussing the heat resistance index (Ts), it is necessary to add references that demonstrate the claim.

Response: Thank you for your comments.References have been added to the article now ‘H. H. Horowitz and G. Metzger, Anal. Chem., 1963, 35, 1464-&.

Point6: The Equation and Ts values should be in the article and not as supplementary material, as it is an important discussion of the work.

Response: Thank you for your comments. It has been revised.

Point7: The calculated value for Ts of the DGEBA is wrong. Consequently, the statement that VAN-AC-EP/DDM has greater thermal stability than DGEBA/DDM is invalid.

Response: Thanks for pointing out the error, it has been corrected

Point8: Page 6, sentence: “The higher carbon residue rate marked the VAN-AC-EP/DDM have a potential application in flame retardant.”

It is necessary to add references that demonstrate the claim.

Response: Thanks for the reviewer’s suggestion.References have been added to the article now Wang B, Ma S, Li Q, et al. Green Chemistry, 2020, 22, 1275-1290.

Point9: Figure 3d should differentiate the E' and Tan delta curves.

Response: The figure has been reuploaded.

Point10: Table S2 is titled mechanical properties, but the first result is the Tg of the materials. Furthermore, the Tg value in Table S2 differs from the values reported in Table S1. Why are they different values?

Response: Tg146 is tested by DMA, and Tg151 is tested by DSC. There are some deviations between the two testing methods. They have been modified and Tg146 is used uniformly.

Point11: How were the Stiffness (kN/m) values shown in Table S2 obtained?

Response: Stiffness can be obtained by DMA test directly.

Point12: What is the experimental error of tensile tests?

Response: Thanks for the reviewer’s suggestion.Datas has been added Table S1

Sample

Tensile strength (MPa)

Young’s modulus

(MPa)

Elongation at break

(%)

VAN-AC-EP

44±2.2

2685±107

2.4±0.09

DGEBA

38±1.8

1966±117

2.9±0.15

Recycled VAN-AC-EP

29±1.3

2979±148

1.1±0.06

CFRP

122±7.3

6598±290

6.1±0.26

Recycled CFRP

82±4.1

9871±592

1.9±0.07

Point13: Later in the article, the elastic modulus value is mentioned. Are these values obtained from Figure 3c? There is a shoulder at the beginning of the DGEBA sample tensile test curve. What does this behavior refer to, and how was the elastic modulus considered?

Response: The value of the stored modulus is obtained from DMA, and the elastic modulus is calculated from the tensile data, intercepting the stress-strain ratio. It may be that the data exit point is not smooth enough leading to the curve defect.

Point14: On page 7, in the text "Equation 2" the number is incorrect

Response: Thank you for your comments.It has been corrected

Point15: The values of Tr in Table S2 are wrong. And apparently, the correct Er region of the DGEBA sample is not plotted on the DMA plot (Figure 3d).

Response: Thanks for the reviewer’s suggestion. The values of Tr has been corrected. The unit of Tr is K and the unit of Tg is oC. Because the value of Er barely changes around Tr, it is not drawn. Now, the figure has been reuploaded.

Sample

Tg

(oC)

 Er

(MPa)

Tr 

(K)

Ve

 (mol/m3)

Stiffness (kN/m)

VAN-AC-EP

146 

6.1

459

568

588

DGEBA

202

26.6

519

2078

303

Point16: On page 8, there is mention of Figure S3. However, there is no Figure S3. Possibly this is Figure 4b.

Response: Thank you for your comments. It has been revised

Point17: On page 8, the third sentence statement needs further discussion and references.

Response: The sentense has been revised The apparent relaxation of the resin occurred, due to the rearrangement of imine bonds. The dynamic exchange of amine bonds gave the reprocessing ability of the thermosetting resin.

Point18: Figure 5a needs to discuss the main vibrations of interest and demonstrate them graphically in the spectra.

Response: Figure 5a shows that the composition of the recycled VAN-AC-EP/DDM had not changed compared with the original VAN-AC-EP/DDM. The stretching vibration peak (1580cm-1) belonging to C=N was also retained.

Point19: What are the dimensions of the recycled VAN-AC-EP/DDM specimen for tensile testing, and what is the experimental error of the analysis?

Response: VAN-AC-EP was ground into powder, evenly spread in a steel groove 70 mm (L) ´ 5 mm (W) ´ 0.5 mm (T) with a polytetrafluoroethylene plate above and below, and placed in a plate vulcanizing machine for hot pressing 180 oC for 0.5h.Recycled film can be obtained by demoulding after cooling at room temperature. And experimental error has been added in Table S1.

Point20: Figure S4 does not have a volume ratio of the acetone/HCL scale.

Response: Thank you for your comments. The figure has been reuploaded.

Point21: Figure 7, what are the legends of the DSC curves?

Response:

Point22: In the conclusions, a third Tg value is reported, different from the other two in Tables S1 and S2. Why the different values? The value of the modulus of elasticity was not discussed throughout the text.

Response: Thanks for the comment. Sorry, It is a mistake in my draft. The wrong value was entered.The elastic modulus is also supplemented.

Reviewer 2 Report

See the attached report please.

Author Response

Response to Reviewers

Thank you very much for your valuable comments. We have revised the manuscript thoroughly according to the reviewers’ recommendations.

For Reviewer #2:

Point1: Bold sentences:

In the introduction section, the sentence “These traditional DGEBA epoxy resins are derived from fossil resources, which makes them non-degradable and non- reprocessing composites.”, is a bold and false sentence. The fact that makes DGEBD non-reprocessing is not its origin, but the existence of a higher crosslink degree after cured, and this is not dependent on the origin of the raw material but on its chemical structure. Please, be precise in your assertions.

Response: Thank you for your comments. I made a mistake and It has been revised in the article. Permanet cross-linked networks formed within the resins makes them non-degradable and non-reprocessing composites.

Point2: In the same section, the authors claim that the incineration destroyed the CFs. Well, this is not always true. If the process is performed under controlled pyrolysis conditions it is not only possible but has been demonstrated to obtain recycled CFs with even much better properties than those reported by the authors in this article. So, be precise. There are plenty of works in the literature on the topic, and even these kinds of processes are being exploited in industry.

Response: Thanks for the reviewer’s suggestion. We have modified the sentence in the article Recycling by landfill incineration causes a huge burden on the environment,these metheds may reduce the quality of carbon fiber and greatly reduce the commercial value of carbon fiber..

Point3: Lack of traceability:

In general, many of the sentences in the introduction section must include some citations. For instance, when introducing CAN, the authors mention its low mechanical properties and so on, but fail in providing a citation to support this assertion. There are other many examples, so, revise your citations.

Response: It has been revised in the article.

Point4: In the materials section, some properties of the materials must be informed. In the materials section, none is said about the CF properties and suppliers.

Response: According to the reviewer’s suggestion, Carbon fiber (T300-3k) was purchased locally.has been added.

Point5: Poor description of the preparation of the CFRR, without indicating the sizing of the CF and if some apparatus or not are used to prepare the composite. The description seems very vague to obtain conclusions and is not traceable for any reader.

Response: Thanks for the reviewer’s suggestion. No special equipment is used to prepare CFRP, the section has been revised as follow:

2.5. Preparation of Carbon Fiber Reinforced Polymer Composites (CFRP)

Firstly, VAN-AC-EP and DDM were fully melted and mixed at 90 oC,and 50wt% ethanol was added to make a mixed solution. The mixture was evenly brushed on carbon fiber (100mm ´100mm). Then the carbon fiber prepreg was obtained by removing ethanol in a vacuum oven at 60 oC for 10 h.Then the carbon fiber composites were perpared by hot pressing at 110 oC, 130 oC and 150 oC for 2 h, respectively.

Point6: The reprocessing procedures of the VAN-AC-EP/DDM are missing; these are not described in the experimental part.

Response: Thank you for your comments.The section has been added as follow:

2.4. Reprocessing of Carbon Fiber Reinforced Polymer Composites (CFRP) Broken parts of CFRP were overlapped, sandwiczed between two polytetrafluoroethyene plate. Hot pressing 180 oC for 0.5 h.After cooling ,the recycled CFRP could be obtained.

Point7: The characterization procedures are sometimes partially described. For instance, since the number of samples influences the DSC and TGA results, this must be informed. The number of tested samples per item must be also included.

Response: Sorry to make you have such misunderstanding. We had conducted a confirmative experiment to make sure that the result of DSC and TGA is correct.

Point8: Equally, the tensile experiment is not completely defined since the authors avoid the number of samples, the use of an extensometer or not, the absence of standard deviation, the use of an excessive rate for modulus determination, and so on.

Response: Thanks for the reviewer’s suggestion. Attention has been paid to the deviation between the storage modulus and the elastic modulus, and the elastic modulus and error have been recalculated.And the tensile data and errors were summarized in Table S1

Point9: The characterization procedure for the recovered CF is not defined in any place.

Response: Thank you for your comments.The section has been added as follow:

2.6. Reprocessing of Carbon Fiber Reinforced Polymer Composites (CFRP)

Broken parts of CFRP were overlapped, sandwiczed between two polytetrafluoroethyene plate. Hot pressing 180 oC for 0.5 h.After cooling ,the recycled CFRP could be obtained.

Point10: Every equation used in the article requires a citation. Note that you have not included references to them either in the article body or in the supplementary information provided.

Response: Thanks for the comment. It has been revised.

Point11: The images in Figure 1 (a, and b) are excessively tiny to be followed the interpretation by any reader. So, please, provide more detailed plots.

Response: Thanks for the reviewer’s suggestion. The figure has been reuploaded.

Point12: This reviewer wonder for the reason for not including information about the loss modulus of the new material, since these spectra would inform much more about the possible more viscous character of VAN-AC-EP respecting DEGBA.

Response: Thanks for the comment. The loss modulus of VAN-AC-EP was 74 MPa while the loss modulus of DEGBA was 65 MPa. We do not described it too much because the resin behaves as a rigid material after cured and it is of little significance to discuss viscosity.

Point12: The dynamic mechanical analysis must be much better exploited since it provides information about the chemical makeup of the material concerning relaxation phenomena. None is said in the text.

Response: We have modified the sentence in the article to make the expression more clearly as follows:

The apparent relaxation of the resin occurred, due to the rearrangement of imine bonds. The dynamic exchange of amine bonds gave the reprocessing ability of the thermosetting resin. The value of Ea is noteworthy,which is much lower than some other thermosetting epoxy resin without the addition of catalysts.This makes resin more malleable at high temperture.

Point12: This reviewer wonders about how different the value of the Storage Modulus respecting the Young Modulus obtained by tensile properties.

Response:

The storage modulus of VAN-AC-EP/DDM was 3350 MPa.

The storage modulus of DGEBA/DDM was 2320 MPa. 

The Young Modulus of VAN-AC-EP/DDM was 2685 MPa.

The Young Modulus of DGEBA/DDM was 1966 MPa.

Point12: This reviewer wonders about the value of the modulus of the Original EP and Recycled EP. They simply mention that are similar. Please, provide the data.

Response: The section has been revised as follow:The Young's modulus of recycled resin was 2979 MPa which almost unchanged and even higher than the original value (2685MPa). 

Reviewer 3 Report

Dear respected author

The present work includes experiment studies on parameters influencing epoxy resin with high performance and the effective recycling of carbon fiber. This is an exciting topic and may improve the knowledge in this regard. However, several points to be considered by the author may improve the level of the manuscript significantly. Such points are addressed as follows before acceptance:

1.       In Abstract:  Important results should be given. % are needed.

2.       When discussing the present work, use the (active or passive) present perfect or simple past tenses. Please, check the manuscript thoroughly.

3.       The introduction section should be improved.After mentioning the objectives of the study in "Introduction" section, please provide significance of this study in engineering sector. Besides, the importance of using composite materials should be included using followings:

Strengthening of shear-critical reinforced concrete T-beams with anchored and non-anchored GFRP fabrics applications Numerical study; Shear strengthening of reinforced concrete T-beams with anchored and non-anchored CFRP fabrics; 

Optimum amount of CFRP for strengthening shear deficient reinforced concrete beams;

4.       There are many grammar mistakes. It should be revised.

5.       Provide a view of the tests used in the considered experiment data.

6.       Estimation of stress of samples with some theoretical approach would be highly appreciated to well validate the experimental analysis.

7.       Provided discussions on the results is not enough. Please discuss the ductility ratio, energy absorption, failure modes etc. in text to better understand the findings.

8.       Please give more detailing for the FRP. Also, please give more detail on epoxy.

9.       Please re-upload all figures at least 300 dpi.

10.   Include a table summarizing all test results and damages.

11.   Coupon test properties of materials should given with light of codes.

12.   The conclusion section needs to be re-written by incorporating general conclusions from the findings of this research. Please point out the novelty of this research article in this section. % are needed.

13.   What is original, and what is significant about this work? Deep discussion are needed.

14.   What is the lesson learned? Please clarify before acceptance.

Author Response

Response to Reviewers

Thank you very much for your valuable comments. We have revised the manuscript thoroughly according to the reviewers’ recommendations.

For Reviewer #3:

Point1: In Abstract: Important results should be given.% are needed.

Response: Thank you for your comments. The sentence has been added This bio-based epoxy resin showed Young’s modulus of 2.68 GPa and tensile strength of 44 MPa, 36.8% and 15.8% higher than those of bisphenol A epoxy respectively..

Point2: When discussing the present work, use the (active or passive) present perfect or simple past tenses. Please, check the manuscript thoroughly.

Response: We have checked the grammar and sentence structure thoroughly.

Point3: The introduction section should be improved.After mentioning the objectives of the study in "Introduction" section, please provide significance of this study in engineering sector. Besides, the importance of using composite materials should be included using followings:

Strengthening of shear-critical reinforced concrete T-beams with anchored and non-anchored GFRP fabrics applications Numerical study; Shear strengthening of reinforced concrete T-beams with anchored and non-anchored CFRP fabrics;

Optimum amount of CFRP for strengthening shear deficient reinforced concrete beams;

Response: The section has been revised Improving the recovery efficiency of thermosetting resins and CFRP composites has become an urgent problem. In order to replace traditional thermosetting resinsin this paper, we aim to fabricate a bio-based epoxy resin with high mechanical property which could be comparable to DGEBA, as well as with well degradable and easy recoverable properties without any other catalysts.

Point4: There are many grammar mistakes. It should be revised.

Response: We have checked the grammar.

Point5: Provide a view of the tests used in the considered experiment data.

Response:

Point6: Estimation of stress of samples with some theoretical approach would be highly appreciated to well validate the experimental analysis.

Response: Thank you for your comments. We'll use this method to verify correctness later on.

Point7: Provided discussions on the results is not enough. Please discuss the ductility ratio, energy absorption, failure modes etc. in text to better understand the findings.

Round 2

Reviewer 1 Report

In Figure 7, indicate which day represents each curve.

Author Response

Thank you very much for your valuable comments. We have revised the manuscript

Reviewer 2 Report

see the attached file.

Author Response

Point4: In the materials section, some properties of the materials must be informed. In the materials section, none is said about the CF properties and suppliers.

Response: According to the reviewers suggestion, ‘Carbon fiber (T300-3k) was

purchased locally.’has been added. Second Draft Reviewer Comments:

I am sorry, but to say that your CF is T300-3k is giving very low informationas to be traceable your experimentation. The authors must note that under this denomination (T300-3k), you can acquire at least four types of CF depending of the twist and the sizing provided. In any case, the inclusion of some properties is also mandatory to give traceability to the results (the realm of any scientific communication).

Response: ‘Carbon fiber (T300-3k, 240 g/m2) was purchased locally.’has been added.

Point5: Poor description of the preparation of the CFRR, without indicating the sizing of the CF and if some apparatus or not are used to prepare the composite. The description seems very vague to obtain conclusions and is not traceable for any reader.

Response: Thanks for the reviewers suggestion. No special equipment is used to

prepare CFRPthe section has been revised as follow:

2.5. Preparation of Carbon Fiber Reinforced Polymer Composites (CFRP)

Firstly, VAN-AC-EP and DDM were fully melted and mixed at 90 oC,and 50wt% ethanol was added to make a mixed solution. The mixture was evenly brushed on carbon fiber (100mm ´100mm). Then the carbon fiber prepreg was obtained by removing ethanol in a vacuum oven at 60 oC for 10 h.Then the carbon fiber composites were perpared by hot pressing at 110 oC, 130 oC and 150 oC for 2 h, respectively.

Second Draft Reviewer Comments:

The sizing of the CF used is not informed (as mentioned in the previous point 4).

Response: The CF specifications have been added in point 4

Point7: The characterization procedures are sometimes partially described. For instance, since the number of samples influences the DSC and TGA results, this must be informed. The number of tested samples per item must be also included. Response: Sorry to make you have such misunderstanding. We had conducted a confirmative experiment to make sure that the result of DSC and TGA is correct.

Second Draft Reviewer Comments:

I am afraid that the authors have not understood my query. This reviewer was answering for the amount and number of samples used in the primary experimentation, and not if you have confirmed of not this value in an additional experiment (that in any case, the authors do not exhibit).

Response: The sample mass of DSC and TGA was 3-5mg, and the experiment was repeated twice for each data to ensure that the results were not accidental.

Point8: Equally, the tensile experiment is not completely defined since the authors avoid the number of samples, the use of an extensometer or not, the absence of standard deviation, the use of an excessive rate for modulus determination, and so on.

Response: Thanks for the reviewers suggestion. Attention has been paid to the deviation between the storage modulus and the elastic modulus, and the elastic modulus and error have been recalculated. And the tensile data and errors were summarized in Table S1

Second Draft Reviewer Comments:

This is still not defined. The authors have not answered each one of the questions in the first draft revision.

Response: Didnt use  extensometer.

The sentence Tensile properties of VAN-AC-EP-DDM and CFRP at room temperature were tested on a 5967X universal testing machine. Size of the resin for tensile tests was 30 mm (L) ´ 5 mm (W) ´ 2 mm (T) and the stretching rate was 10 mm/min. The distance between two clamps was 10 mm. Each tensile data was obtained from an average of 5 samples. has been added

Point10: Every equation used in the article requires a citation. Note that you have not included references to them either in the article body or in the supplementary information provided.

Response: Thanks for the comment. It has been revised. Second Draft Reviewer Comments:

The authors claim that it has been revised, but they have forgotten to include the information in some places. The authors have included this information for equations 3, 5, and 6 but not for the others. Please, include them.

Response: They has been added.

Point12: This reviewer wonder for the reason for not including information about the loss modulus of the new material, since these spectra would inform much more about the possible more viscous character of VAN-AC-EP respecting DEGBA.

Response: Thanks for the comment. The loss modulus of VAN-AC-EP was 74 MPa while the loss modulus of DEGBA was 65 MPa. We do not described it

too much because the resin behaves as a rigid material after cured and it is of

little significance to discuss viscosity.

Second Draft Reviewer Comments:

This reviewer still thinks that it is interesting to include this information,

mainly due to the so different Tg found between the compared compounds. In

their response, the value provides the loss modulus at a single point, without

indicating what T corresponds. This reviewer must insist is visualizing these

spectra with the evolution of loss modulus with T, at least in the

supplementary material section.

Response: Loss modulus curves has been added (Figure S3).

Point12: The dynamic mechanical analysis must be much better exploited

since it provides information about the chemical makeup of the material

concerning relaxation phenomena. None is said in the text.

Response: We have modified the sentence in the article to make the expression

more clearly as follows:

‘The apparent relaxation of the resin occurred, due to the rearrangement of

imine bonds. The dynamic exchange of amine bonds gave the reprocessing

ability of the thermosetting resin. The value of Ea is noteworthy, which is

much lower than some other thermosetting epoxy resin without the addition

of catalysts. This makes resin more malleable at high temperture.’

Second Draft Reviewer Comments:

The sentence included makes even more sense to claim the inclusion of the

loss modulus evolution. This reviewer wants to see how this more malleable

effect is observed in the viscous behavior of the material.

Response: ‘malleable’ here means flexible and easy to change shape.When the temperature is above Tg, the shape of the resin can be easily changed. Ea represents the activation energy, and theoretically the lower the value, the more plastic the resin will be.

(Roig A , A Petrauskaité, Ramis X , et al. Synthesis and characterization of new bio-based poly(acylhydrazone) vanillin vitrimers[J]. Polymer Chemistry.)

Point12: This reviewer wonders about how different the value of the Storage

Modulus respecting the Young Modulus obtained by tensile properties.

Response:

The storage modulus of VAN-AC-EP/DDM was 3350 MPa.The storage modulus of DGEBA/DDM was 2320 MPa.

The Young Modulus of VAN-AC-EP/DDM was 2685 MPa.

The Young Modulus of DGEBA/DDM was 1966 MPa.

Second Draft Reviewer Comments:

Without providing the standard deviation, these values appear as bold. At a

glance, Young Modulus and Storage Modulus give similar values (as expected).

So, if both parameters provide the same information, this reviewer wonders if

the authors can explain the use of both parameters instead of just the coming

from the tensile test, or the coming from the DMA test. Please, provide a

justified explanation for this fact.

Response:

The stress-strain curves showed that the tensile strength of VAN-AC-EP/DDM reached 44 MPa, higher than that of DGEBA/DDM (38 MPa). While the elongation at break was 2.4%, slightly lower than that of DGEBA/DDM system and the young modulus of VAN-AC-EP/DDM was 2685 MPa was much higher than it of DGEBA/DDM (2320 MPa). The higher mechanical strength of VAN-AC-EP/DDM was provided by a large number of the C=N double bonds in the molecular backbone of the VAN-AC-EP/DDM system after curing and π-π conjugation existed between the benzene ring and the imine bond. The higher molecular rigidity limited the rotation of the molecular chains, which induced the decrement of the flexibility.

Point12: This reviewer wonders about the value of the modulus of the

Original EP and Recycled EP. They simply mention that are similar. Please,

provide the data.

Response: The section has been revised as follow: ’The Young's modulus of

recycled resin was 2979 MPa which almost unchanged and even higher than

the original value (2685MPa).’

Second Draft Reviewer Comments:

Attending the Standard deviation, the modulus rather overlaps. Note that you

have used an excessive testing rate for modulus, typically of 1mm/min, to

affirm that the two you have compared are different.

Response: We don't think the stretching rate is going to make much difference,In some literature, the tensile rate is 20mm/min.

Wang B, Ma S, Li Q, et al. Green Chemistry, 2020, 22, 1275-1290 

Wang S , Xing X , Zhang X , et al. Room-temperature fully recyclable carbon fibre reinforced phenolic composites through dynamic covalent boronic ester bonds[J].

Journal of Materials Chemistry A, 2018:10.1039.C8TA01801D.

Sustainable Epoxy Vitrimers from Epoxidized Soybean Oil and Vanillin[J]. ACS Sustainable Chemistry And Engineering, 2020, 8(39):15020-15029.

Reviewer 3 Report

The manuscript may be accepted in its current form.

Author Response

Thank you for your review

Round 3

Reviewer 2 Report

See attached file.

Author Response

Response to Reviewers

Thank you very much for your valuable comments. We have revised the manuscript thoroughly according to the reviewers’ recommendations.

For Reviewer #2:

Point4:

In the materials section, some properties of the materials must be informed. In the materials section, none is said about the CF properties and suppliers.

Response:

                                                          -3k) was purchased

Second Draft Reviewer Comments:

I am sorry, but to say that your CF is T300-3k is giving very low information as to be traceable your experimentation. The authors must note that under this denomination (T300-3k), you can acquire at least four types of CF depending of the twist and the sizing provided. In any case, the inclusion of some properties is also mandatory to give traceability to the results (the realm of any scientific communication).

Response:

-

Third Draft Reviewer Comments:

This question is not solved yet. This reviewer just claim to know if the T300-3k that authors have used is Twist A, or Twist B type, and if the sizing in the called 40A, 40B, 50A or 50B, just so. The authors has includes a label in their answer to this concern but it is written in Chinese, and (unfotunatelly) this reviewer cannot read Chinese.

Response:

Carbon fiber (T300-3k, 240 g/m2, 50C) was purchased locally. 

Not twist.

Point 5:

Poor description of the preparation of the CFRR, without indicating thesizing of the CF and if some apparatus or not are used to prepare the composite. The description seems very vague to obtain conclusions and is not traceable for any reader.

Response:

prepare CFRP

the section has been revised as follow:

2.5. Preparation of Carbon Fiber Reinforced Polymer Composites (CFRP).

Firstly, VAN-AC-EP and DDM were fully melted and mixed at 90 oC,and 50wt% ethanol was added to make a mixed solution. The mixture was evenly brushed on carbon fiber

in a vacuum oven at 60oC for 10 h. Then the carbon fiber composites were prepared by hot pressing at 110 oC, 130oC and 150oC for 2 h, respectively.

Second Draft Reviewer Comments:

The sizing of the CF used is not informed (as mentioned in the previous point 4). Response:

The CF specifications have been added in point 4

Third Draft Reviewer Comments:

See answer to point 4. The only information that this reviewer require is to know if the sizing is 40A, 40B, 50A, or 50B. Note that T300-3k offers all these options.

Response: The CF specifications have been added in point 4

Point 12:

This reviewer wonders about how different the value of the Storage Modulus respecting the Young Modulus obtained by tensile properties.

Response:

The storage modulus of VAN-AC-EP/DDM was 3350 MPa.The storage modulus of DGEBA/DDM was 2320 MPa. The Young Modulus of VAN-AC-EP/DDM was 2685 MPa. The Young Modulus of DGEBA/DDM was 1966 MPa.

Second Draft Reviewer Comments:

Without providing the standard deviation, these values appear as bold. At a glance, Young Modulus and Storage Modulus give similar values (as expected). So, if both parameters provide the same information, this reviewer wonders if the authors can explain the use of both parameters instead of just the coming from the tensile test, or the coming from the DMA test. Please, provide a justified explanation for this fact.

Response:

The stress-strain curves showed that the tensile strength of VAN-AC-EP/DDM reached 44 MPa, higher than that of DGEBA/DDM (38 MPa). While the elongation at break was 2.4%, slightly lower than that of DGEBA/DDM system and the young modulus of VAN- AC-EP/DDM was 2685 MPa was much higher than it of DGEBA/DDM (2320 MPa). The higher mechanical strength of VAN-AC-EP/DDM was provided by a large number of the

C=N double bonds in the molecular backbone of the VAN-AC-EP/DDM system after

- existed between the benzene ring and the imine bond. The higher molecular rigidity limited the rotation of the molecular chains, which induced the decrement of the flexibility.

Third Draft Reviewer Comments:

This reviewer is afraid that the authors have not understood my previous question. The explanation given by the authors (although very interesting) is not related with this.

The only that this reviewer wants to know if why the authors reported young modulus and storage modulus when both of them offer almost the same information on the elastic behavior of the material, just so. Please, provide and explanation.

Response: The storage modulus reflects the storage modulus of the material due to elastic deformation, and the elastic modulus indicates the resistance capacity of the material to deformation. Although the data measured by these two moduli are similar, there are still differences.Both of these modulus are typical key data of mechanical properties and can be analyzed together

In light of the above-mentioned unsolved concerns, the recommendation must be to perform a minor revision. Note that they are extremely easy to be solved.
